# Contrastive Modules with Temporal Attention for Multi-Task Reinforcement Learning

**Siming Lan[1,2,3]**   **Rui Zhang[2]**   **Qi Yi[1,2,3]**   **Jiaming Guo[2]**   **Shaohui Peng[5]**
**Yunkai Gao[1,2,3]**   **Fan Wu[2,3,4,5]**   **Ruizhi Chen[5]**   **Zidong Du[2,6]**   **Xing Hu[2,6]**
**Xishan Zhang[2,3]**   **Ling Li[4,5]**   **Yunji Chen[2,4] †**

[1] University of Science and Technology of China
[2] State Key Lab of Processors, Institute of Computing Technology,
Chinese Academy of Sciences, Beijing, China
[3] Cambricon Technologies
[4] University of Chinese Academy of Sciences, UCAS, Beijing, China
[5] Intelligent Software Research Center, Institute of Software, CAS, Beijing, China
[6] Shanghai Innovation Center for Processor Technologies, SHIC, Shanghai, China
{lansm,gyk314,yiqi}@mail.ustc.edu.cn
{zhangrui,guojiaming,duzidong,huxing,zhangxishan,cyj}@ict.ac.cn
{pengshaohui,wufan2020,ruizhi,liling}@iscas.ac.cn

## Abstract

In the field of multi-task reinforcement learning, the modular principle, which involves specializing functionalities into different modules and combining them appropriately, has been widely adopted as a promising approach to prevent the negative transfer problem that performance degradation due to conflicts between tasks. However, most of the existing multi-task RL methods only combine shared modules at the task level, ignoring that there may be conflicts within the task. In addition, these methods do not take into account that without constraints, some modules may learn similar functions, resulting in restricting the model's expressiveness and generalization capability of modular methods. In this paper, we propose the Contrastive Modules with Temporal Attention(CMTA) method to address these limitations. CMTA constrains the modules to be different from each other by contrastive learning and combining shared modules at a finer granularity than the task level with temporal attention, alleviating the negative transfer within the task and improving the generalization ability and the performance for multi-task RL. We conducted the experiment on Meta-World, a multi-task RL benchmark containing various robotics manipulation tasks. Experimental results show that CMTA outperforms learning each task individually for the first time and achieves substantial performance improvements over the baselines. Our code can be found at https://github.com/niiceMing/CMTA.

## 1   Introduction

Though deep reinforcement learning (RL) has made significant progress in multiple domains such as playing Atari games [22] and robotic control[16], most of these methods tackle different tasks in isolation, making it difficult to leverage the previously learned skills for a new task. Multi-task RL aims to train multiple tasks simultaneously in a sample efficient manner [6] by using shared

---

† Corresponding author.

37th Conference on Neural Information Processing Systems (NeurIPS 2023).

and re-used components across different tasks. Since different tasks may have complementary information or act as a regularizer for each other, multi-task RL methods have the potential to get better performance and generalization [4].

There is a vital challenge in multi-task RL if two tasks are essentially unrelated or have conflicts, increasing the performance of the method on one task will hurt the performance on the other, referred as *negative transfer* problem [42, 37]. The occurrence of negative transfer is attributed to the utilization of the same model to learning different tasks or different functions. When conflicts arise, the model needs to compromise among multiple tasks, which often results in inferior performance compared to training each task individually. To tackle this problem, researchers propose to exploit the task relationships [35, 34] along with basic shared models, including task-specific decoding head[17], routing networks [10, 29], cross-stitch networks [21], compositional modules[8, 44, 2] and mixture of experts[9, 20, 34]. These methods use different combinations of shared modules or different parts of the shared model according to different tasks, which can be seen as an embodiment of modular approaches.

The main idea of modular methods is to separate and specialize the functionalities into *different modules*, allowing them to focus on their respective skills or features without interference or conflicting objectives. Different modules can collaborate and complement each other's strengths. By *appropriately combining* these distinct modules, it is theoretically possible to resolve the trade-off dilemma encountered or negative transfer in multi-task RL and enhance the model's expressive power and generalization ability. And it has been observed in neuroscience research that different regions of the brain possess completely different functions, aligning with the concept of modularity. This may be one of the reasons why humans are capable of rapidly learning multiple tasks.

Despite the widespread adoption of modular principle in most multi-task RL methods, these methods still underperform compared to single-task training. We argue that the inferior performance of these works may be attributed to the fact that they have not fully adhered to the principle of modularity, which hinders their ability to mitigate negative transfer. One of the key aspects of modularity is ensuring that *different modules* specialize in distinct functionalities, but these methods only partition the network into multiple modules in a superficial way without further constraints which may result in some modules learning the same functionality. This goes against the modular principle and restricts the model's expressiveness and generalization capability of modular methods and leads to redundant computations and even overfitting. Another challenge lies in the *appropriate combination* of modules. Most methods only combine shared modules at the task level, using a fixed combination of modules for each task during the whole episode. However, negative transfer exists not only between tasks but also within tasks. For instance, considering the task 'Push' in Meta-World(Figure 1), the agent needs to reach the puck at first and then push it to a goal, these two purposes are not identical and not always be mutually supportive, so negative transfer may exist when using a same module combination at these time steps.

In this paper, we propose a novel approach called Contrastive Modules with Temporal Attention (CMTA) to address the limitations of existing multi-task RL methods and further alleviate negative transfer in multi-task RL. The key insight lies in constraining the modules to be different from each other by contrastive learning and combining shared modules at a finer granularity than the task level with temporal attention. Roughly speaking, CMTA uses LSTM to encode the temporal information, along with a task encoder to encode the task information. Then CMTA takes both temporal and task information to compute the soft attention weights for the outputs(encodings) of modules, and we apply contrastive loss to them, aiming to increase the distance between different encodings. Therefore, CMTA enforces the distinctiveness of modules and provides the agent with a finer-grained mechanism to dynamically select module combinations within an episode, alleviating the negative transfer within the task and improving the generalization ability and the performance for multi-task RL.

We conduct experiments on Meta-World[48], a multi-task RL benchmark containing 50 robotics manipulation tasks. Experimental results show that the proposed CMTA significantly outperforms all baselines, both on sample efficiency and performance, and even gets better performance than learning each task individually for the first time. CMTA also shows great superiority especially for complex tasks. As the variety of tasks increases, the advantages of CMTA become more apparent.

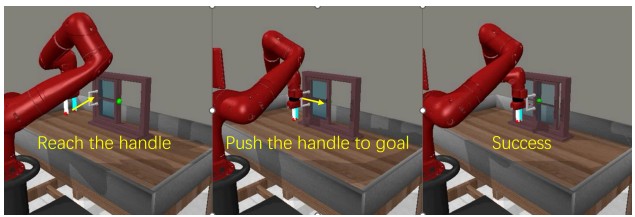

Figure 1: Different purpose in Meta-World manipulation task 'Push' : Push the puck to a goal.

## 2    Related work

**Multitask Learning.** Multi-task learning (MTL) can learn commonalities and differences across different tasks. [4, 6, 7, 41, 30]. Researchers have shown MTL can make multiple tasks benefit from each other by sharing useful information and exploiting the commonality and structure of these tasks [18, 28, 19, 27]. [49] shows that different tasks have different similarities, and some unrelated tasks may have negative impact on each other when training together. A large portion of the MTL work is devoted to the design of multi-task neural network architectures that can avoid negative transfer, [31, 17] use a shared encoder that branches out into task-specific decoding heads. [21] proposed Cross-stitch networks, which softly share their features among tasks, by using a linear combination of the activations found in multiple single task networks. Multi-Task Attention Network [19] consists of a single shared network containing a global feature pool, together with a soft-attention module for each task.

**Compositional modules and Mixture of experts.**  In order to reduce the negative impact of gradient conflict, a feasible method is to use compositional modules, which makes the model have better generalization. [3] leverages the compositional structure of questions in natural language to train and deploy modules specifically catered for the individual parts of a question. [8] decomposes policies into "task-specific" (shared across robots) and "robot-specific" (shared across tasks) modules, and the policy is able to solve unseen tasks by re-combining the pre-defined modules. Routing Networks [29] is comprised of a router and a set of neural network modules, and the composition of modules is learned in addition to the weights of the modules themselves. Instead of directly selecting routes for each task like Routing Networks[29], soft modularization[44] uses task-specific policy softly combine all the possible routes.

In the deep learning era, ensemble models and ensemble of sub-networks have been proven to be able to improve model performance[38, 15]. [9]turn the mixture-of-experts model into basic building blocks (MoE layer). The MoE layer selects experts based on the input of the layer at both training time and serving time. [23] selects one specific expert at a time for each input. [32] use a gating function to select experts. [20] use multi-gate method, and introduce a gating network for each task.

**Task embedding.**  Task embedding is a very general form of learning task relationships, and has been widely used in meta-learning, as well as multi-task learning. It uses task information to model the explicit relationships between tasks, such as clustering tasks into groups by similarity, and leveraging the learned task relationships to improve learning on the tasks at hand. [1] uses task embedding to generate vectorial representations of visual classification tasks which can be used to reason about the nature of those tasks and their relations. [14] uses metric learning to create a task embedding that can be used by a robot to learn new tasks from one or more demonstrations. [34] uses metadata, a pre-trained task embedding obtained by task description and a pre-trained language model, to help the robot make use of the relation between continuous control tasks.

**Contrastive Learning.**  Contrastive Learning is a framework that aims to learn representations that adhere to similarity constraints in a dataset organized into pairs of similar and dissimilar items. This can be thought of as performing a dictionary lookup task, where the positive and negative pairs represent a set of keys relative to a query (or anchor). One of the most basic forms of contrastive learning is Instance Discrimination [43], in which a query and a key are considered positive pairs if they are data augmentations of the same instance (e.g. image) and negative pairs otherwise. However, choosing appropriate negative pairs can be a significant challenge in contrastive learning, as it can greatly impact the quality of the learned representations. There are several loss functions available for use in contrastive learning, such as InfoNCE [39], Triplet[36], Siamese [5], and others.

# 3 Preliminaries

**Markov Decision Process (MDP)** is defined by a tuple $(S, A, P, R, \gamma)$ where $S$ is a finite set of states, $A$ is a finite set of actions, transition function $P(s'|s, a)$ is a transition probability from state $s$ to state $s'$ after action a is executed, reward function $R(s, a)$ is the immediate reward obtained after action $a$ is performed, and discount factor $\gamma \in (0, 1)$. We denote $\pi(a|s)$ as the policy which is a mapping from a state to an action. The goal of an MDP is to find an optimal policy to maximize the reward function. More specifically, in finite horizon MDP, the objective is to maximize the expected discounted total reward which is defined by $max_\pi E_\tau[\sum_{t=0}^T \gamma^t R(s_t, a_t)]$, where $\tau$ is trajectory following $a_t \sim \pi(a_t|s_t)$, $s_{t+1} \sim P(s_{t+1}|s_t, a_t)$. The value function of policy $\pi$ is defined as $V_\pi(s_t) = E_\pi[\sum_{t=0}^T \gamma^t R(s_t, a_t)]$.

**Multi-task RL optimization.** In general multi-task learning with N tasks, we optimize the policy to maximize the average expected return across all tasks, and the total loss function is defined as:

$$L_{total} = \sum_{i=1}^N \lambda_i L_i. \tag{1}$$

This is the linear combination of task-specific losses $L_i$ with task weightings $\lambda_i$.

**Contrastive learning.** Given a query $q$, positive keys $k^+$ and negative keys $k^-$, the goal of contrastive learning is to ensure that $q$ matches with $k^+$ more than $k^-$. An effective contrastive loss function, called InfoNCE[39], is:

$$L_q = -log \frac{exp(q \cdot k^+/\tau)}{exp(q \cdot k^+/\tau) + \sum_{k^-} exp(q \cdot k^-/\tau)}, \tag{2}$$

where $\tau$ is a temperature hyper-parameter.

# 4 Method

We now introduce our novel multi-task RL architecture, Contrastive Modules with Temporal Attention, which combines different modules with finer granularity by using temporal attention and contrastive modules. CMTA has the ability to alleviate negative transfer within the task, improve performance, enhance generalization, and achieve state-of-the-art results. Notably, it surpasses the performance of learning each task individually for the first time in the Meta-World[2] environment[48].

## 4.1 Contrastive Modules

Employing a modular approach to reuse various modules at different time steps can result in enhanced performance and better generalization. Ensemble models and the mixture of experts have been proven to be able to improve model performance [38, 15] and we use mixture of experts as our shared modules. However, without proper constraints, some modules may redundantly learn the same skill or feature, which goes against the original intention of using the modular approach, limiting the generalization ability of modular methods and only adding computational overhead without improving performance.

To tackle this problem, we introduce contrastive learning to enforce the distinctiveness of modules. Most researchers use contrastive learning to obtain a more robust representation, whereas we use contrastive learning to encourage the modules to be different from each other.

At time step t, for $i$-th encoder's output $z_t^i$ as query $q_i$, we use other encoder's output $z_t^j (j \neq i)$ as its negative pairs $k_i^-$, and the output at the next time step of $i$-th encoder $z_{t+1}^i$ is selected as the positive pair $k_i^+$. The total contrastive loss can be written as:

$$L_{con} = \sum_{i=1}^k -log \frac{exp(q_i \cdot k_i^+/\tau)}{exp(q_i \cdot k_i^+/\tau) + \sum_{k_i^-} exp(q_i \cdot k_i^-/\tau)}, \tag{3}$$

---

[2]This is not the partially observable environment, because the task id is available and can be used to uniquely discriminate the task.

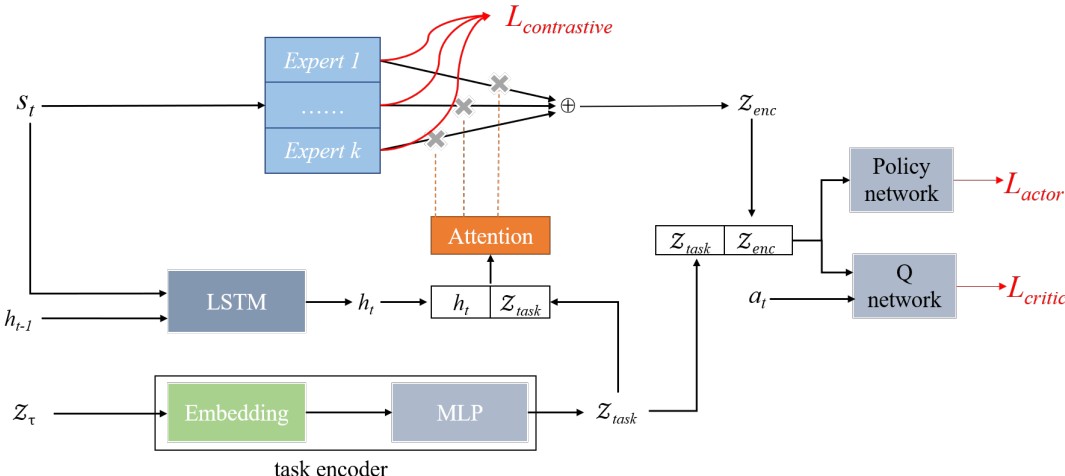

Figure 2: Architecture of our CMTA model. Given the current state $s_t$, we use a mixture of k experts to get k encodings, and extract the task embedding $z_{task}$ by a task encoder using one-hot task id $z_\tau$. And we use LSTM with $h_{t-1}$ and $s_t$ as inputs, output current temporal information $h_t$. The soft attention weights $\alpha$, which is used to compute a weighted sum of multiple encodings, take $h_t$ and $z_{task}$ as inputs.

where $K$ denotes the number of experts.

## 4.2 Temporal attention

Most methods neglect the negative transfer within tasks by only combining shared modules at the task level, using a fixed combination of modules for each task throughout the entire episode. To address this issue, we propose combining the modules with finer granularity at each time step by using temporal attention, further alleviating negative transfer within tasks.

**Mixture of experts.** We use mixture of experts as our shared modules, implemented by k individual encoders. Each representation $z_{enc}^j$ output by the corresponding expert, encodes different features of the state $s_t$, and can be represented as:

$$z_{enc}^j = f^j(s_t), \tag{4}$$

where $f^j$ denotes the $j$-th expert.

**Temporal Attention module.** In order to combine the shared modules dynamically, we use attention module to output the soft combination weights.

Given the current state $s_t$, we encode temporal information $h_t$ as:

$$h_t = lstm(s_t; h_{t-1}), \tag{5}$$

where $lstm$ is a long short-term memory [13], $h_{t-1}$ is previous hidden state at time $t-1$, and $h_0$ is initialized with zero vector.

Given the input one-hot task id $Z_\tau$, the task information can be computed as:

$$z_{task} = g(z_\tau), \tag{6}$$

where $g$ is the task encoder. Notably, [34] uses a pre-trained embedding layer trained by task description and language model, called metadata. They argue that knowing the task description such as 'open a door' and 'open a drawer' is helpful for modeling the task relationships.

By concatenating the temporal information $h_t$ and task information $z_{task}$, we can dynamically compute appropriate attention weights within the episode as:

$$\alpha_1, \cdots, \alpha_k = softmax(\mathcal{W}(z_{task}; h_t)), \tag{7}$$

where $\mathcal{W}$ is a fully connected layer, and $softmax$ is a normalize function which restricts the sum of $\alpha_i$ to 1.

And the final representation is a weighted sum of k multiple representations using soft attention weights $\alpha$ as:

$$z_{enc} = \sum_{j=1}^{k} \alpha_j \cdot z_{enc}^j, \tag{8}$$

which can be seen as a combination of different modules.

## 4.3 Training approach

We use Soft Actor-critic (SAC)[12] to train our policies. SAC is an off-policy actor-critic deep RL algorithm based on the maximum entropy reinforcement learning framework. The optimization objectives of policy and Q-function are:

$$J_Q(\theta) = \underset{\substack{(s,a,s')\sim D, \\ a'\sim\pi_\phi}}{E} [\frac{1}{2}(Q_\theta(s,a) - (r(s,a) + \gamma Q_\theta(s',a'))^2], \tag{9}$$

$$J_\pi(\phi) = \underset{\substack{s\sim D, \\ a\sim\pi_\phi}}{E} [log\pi_\phi(a|s) - \frac{1}{\alpha}Q_\theta(s,a)], \tag{10}$$

where $\alpha$ is a temperature parameter that controls how important the entropy term is. Note that we only use actor loss to update the policy network, and use critic loss to update the whole model. The algorithm is described in Appendix A.

For the multi-task optimization with N tasks, we optimize the policy without any loss weighting tricks, which means we treat each task equally. The total loss is computed as:

$$L_{total} = \sum_{i=1}^{N} \frac{1}{N} L_i, \tag{11}$$

$$L_i = L_{RL} + \beta \cdot L_{con}, \tag{12}$$

where $L_i$ denotes the loss function of task $i$, $\beta$ is a hyper-parameter.

# 5 Experiment

In this section, we evaluate our method on multi-task robotics manipulation. We introduce the environment in Section 5.1, propose an appropriate evaluation metric in Section 5.3 , and compare with baseline methods on Meta-World in Section 5.2 and Section 5.4 and conduct ablation study in Section 5.5.

## 5.1 Environment

We evaluate the effectiveness of our CMTA model on Meta-World environment[48], which is a collection of robotic manipulation tasks designed to encourage research in multi-task RL. We use the mean success rate of the binary-valued success signal as our evaluation metric, which has been clearly defined in Meta-World[48]. Meta-World consists of 50 distinct robotic manipulation tasks with a simulated Sawyer robotic arm, each task has its own parametric variations 50 sets of different initial positions of the object and the goal. The multi-task RL benchmarks within Meta-World are MT10 and MT50, which consist of simultaneously learning 10 and 50 tasks. In the original setting, MT10 and MT50 only use one fixed position for each task during the training stage, which may cause the agent to simply memorize the position and overfitting. To reflect the generalization ability of the model, we mix all 50 sets of positions together for tasks in MT10 and MT50 at training stage, and name them as **MT10-Mixed** and **MT50-Mixed**. The original MT10 and MT50 is denoted as **MT10-Fixed** and **MT50-Fixed**.

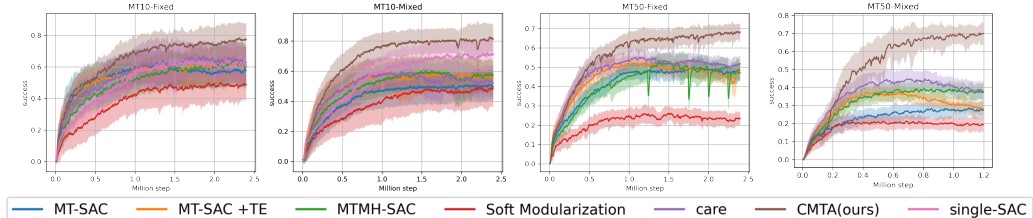

Figure 3: Training curves of different methods on MT10 and MT50 benchmarks, each curve is averaged over 8 seeds(Shaded areas: the standard deviation over 8 seeds). Our approach consistently outperforms baselines in all environments, whether on asymptotic performance or sample efficiency, and the superiority is more obvious on MT10-Mixed and MT50-Mixed. TE = Task Encoder.

## 5.2 Baselines

We take the following multi-task RL methods as the object of our comparison, considered as the baselines: **(i)Single-task SAC(upper bound)**: Using a separate network for each task, which is usually considered as the performance upper bound. **(ii)Multi-task SAC(MT-SAC[48])**: Using a shared network with the state as input. **(iii)Multi-task SAC with task encoder(MT-SAC+TE)**: Adding a task encoder with a one-hot task id as input based on MT-SAC. **(iv)Multi-task multi-head SAC (MTMH-SAC[48])**: Using a shared network with task-specific heads for each task. **(v)Soft Modularization(SoftModu[44])**: It contains several compositional modules as the policy network, and uses a task-specific routing network softly combine the module in the policy network. **(vi)Contextual attention-based representation (CARE[34])**: CARE uses a mixture of encoders, and uses a context encoder with additional prior knowledge metadata to context-based attention.

## 5.3 Implementation details

The experimental results of reinforcement learning environments like Meta-World have great randomness, and the number of seeds plays an important part in the final performance of the model. Specifically, as mentioned in [34], MTMH-SAC achieved a success rate of 88% (1 seed) in [48], which is 44% different from 10seed's performance. In our experiment, each agent is trained over 8 seeds, and we average the mean success rate of each seed to measure the performance. We train the agent with 2.4 million steps (per task) for both MT10 and MT50 environments. More implementation details of hyper-parameters can be found in Appendix D.

During training phase, we evaluate the agent every 3K steps (20 episodes), and we find that the smoothed testing curve and the smoothed training curve are almost identical. It might be the case that the temperature parameter $\alpha$ becomes very small in the training period, which discourages exploration, that is to say, the policy has the same behavior without stochasticity in the training and testing phase.

Even though the testing results have been averaged over 8 seeds, it is still very volatile. Previous work often use the maximum of the mean success rate in evaluation, which is much higher than the average, since the max operation will bring much larger variances. In our experiments with 8 seeds, there is a big gap between the maximum of the mean success rate and the maximum of the smoothed (smooth factor 0.8) mean success rate, which is about $5\% \sim 10\%$, bringing more uncertainty to the evaluation of model performance. Predictably, this uncertainty is much bigger in the case of fewer seeds. [44, 34] tries to reduce uncertainty by evaluating the agent several times at once, but it doesn't really work. Because in the original MT10 and MT50, each task environment has a fixed initial position, and the policy based on SAC is determined during evaluating. In other words, given the parameters of the model, no matter how many times we evaluate the agent, the result is exactly the same as the first time in the testing stage. Thus, compared to the maximum of the mean success rate, we use the *maximum of the smoothed mean success rate* to measure the performance of models, which is more stable and realistic.

Table 1: Evaluation performance on MT10 and MT50 tasks after training for 2.4 million steps (for each task), using the metric of max success rate and max smoothed success rate, where the prior has smaller variance. Results have been averaged over 8 seeds. Our CMTA outperforms other methods on all test environments, including the Single-SAC which is considered as the upper bound. TE = Task Encoder.

| agent | MT10-Fixed success rate | | MT10-Mixed success rate | | MT50-Fixed success rate | | MT50-Mixed success rate | |
| --- | --- | --- | --- | --- | --- | --- | --- | --- |
| | max smoothed | max | max smoothed | max | max smoothed | max | max smoothed | max |
| MT-SAC | 62.25% | 68.75% | 53.22% | 62.50% | 50.37% | 52.50% | 28.78% | 31.50% |
| MT-SAC+TE | 64.76% | 70% | 61.12% | 68.75% | 52.45% | 54.75% | 37.59% | 40% |
| MTMH-SAC | 65.21% | 70% | 62.06% | 67.50% | 47.67% | 48.75% | 39.65% | 42.75% |
| SoftModu | 51% | 55% | 51.34% | 58.75% | 26.23% | 28.75% | 21.50% | 23.50% |
| CARE | 68.03% | 75% | 61.35% | 67.50% | 55.47% | 57.50% | 45.00% | 48.50% |
| CMTA(ours) | **78.95%** | **83.75%** | **82.07%** | **87.5%** | **68.90%** | **71.00%** | **71.69%** | **74.5%** |
| Single-SAC(upper bound) | 64.33% | 68.75% | 71.11% | 76.25% | / | / | / | / |

## 5.4 Results

To evaluate the performance, we compare CMTA with other baselines on MT10-Fixed, MT10-Mixed, MT50-Fixed and MT50-Mixed environments. The training curves averaged over 8 seeds are plotted in Figure 3. And Table 1 records the best evaluation results of all the methods, including both max smoothed (with smooth factor 0.8, details can be found in APPENDIXC) mean success rate and max mean success rate (large variance). It is noteworthy that our method is even better than CARE, which uses additional prior knowledge.

As shown in Figure 3, our approach consistently outperforms baselines in all environments, whether on asymptotic performance or sample efficiency. In the original environments MT10-Fixed and MT50-Fixed, it can be seen from Table 1 that CMTA is better than other baselines around 15% and 24%, respectively. And in the more difficult environment with changing positions, MT10-Mixed and MT50-Mixed, the superiority of our method is more obvious, which is about **34%** and **60%** better than other baselines respectively.

When changing from Fixed environment to Mixed environment, it becomes more difficult because the agent have to deduce patterns or objectives of tasks (e.g., moving an object to a goal) rather than merely memorizing fixed positions. All the baseline methods suffered this impact in MT10-Mixed and MT50-Mixed, but our method achieves even better performance, which implies good generalization and robustness of our method.

Significantly, we notice that only our approach outperforms learning each task individually (we only conduct this experiment on MT10, because it is too slow to train 50 policies simultaneously on MT50). This evidence proves that our approach can avoid negative transfer and improve the performance of tasks by correctly combining the modules. While all the methods can take advantage of multi-task learning to compress the model and improve sample efficiency, in contrast to ours, other methods come at the cost of performance.

As mentioned before, RL experimental result has large stochasticity, and the evaluation metric used in previous works ([48, 44, 34]) is max success rate during testing, which leads to a large variance and is influenced by evaluation times. Thus we use max smoothed success rate to mitigate the effects of large variances caused by max operations. Because of these randomnesses, it is reasonable that the results are somewhat different from the previous works. However, the result of soft modularization[44] in our experiment is much worse than previous work. In addition to the number of seeds and the randomness of the RL algorithm, we guessed that the most likely reason is that SoftModu uses a loss weighting trick in the optimization, which balances the individual loss for different tasks. For the sake of fairness, we don't use any loss weighting method. The ablation study of SoftModu[44] also shows that MTMH-SAC outperforms SoftModu without balance, which is consistent with our experimental results.

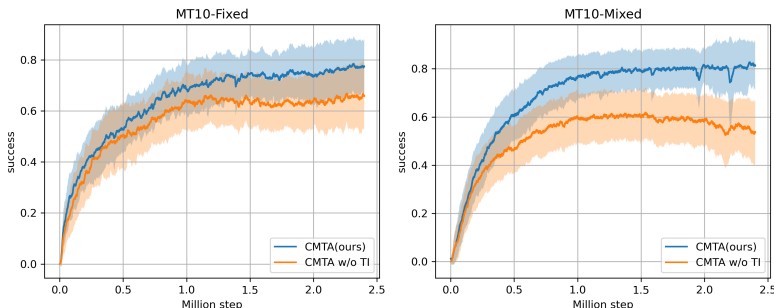

Figure 4: Effectiveness of temporal information(TI) on MT10-Fixed and MT10-Mixed environment, each curve has been averaged over 8 seeds.

Table 2: Effectiveness of contrastive loss(CL) on MT10 and MT50 tasks after training for 2.4 million steps (for each task). Results have been averaged over 8 seeds.

| agent | MT10-Mixed success rate | | MT50-Mixed success rate | |
|---|---|---|---|---|
| | max smoothed | max | max smoothed | max |
| CARE | 61.35% | 67.50% | 45.00% | 48.50% |
| CARE + CL | 65.24% | 71.25% | 47.61% | 49.75% |
| CMTA w/o CL | 79.46% | 85% | 62.66% | 65% |
| CMTA(ours) | **82.07%** | **87.5%** | **71.69%** | **74.5%** |

### 5.5 Ablation Study

**Temporal Attention.** We investigate the significance of temporal information and report the comparison results in Figure 4. It can be seen that the temporal information we employed is indeed highly effective, which indicates that the negative transfer within tasks is further alleviated.

**Contrastive Modules.** We conducted comparative experiments to validate the effectiveness of contrastive loss, as shown in Table 8. From the results, it can be seen that the CL loss consistently improves performance in the mixed environment, especially in MT50-Mixed where our method achieves an increase in success rate of nearly 10 percentage points. Additionally, we observed that the impact of CL loss in the fixed environment is not substantial. This could be attributed to the fixed environment being overly homogeneous, which makes it prone to overfitting and fails to demonstrate the robust generalization advantage of contrastive modules.

To verify whether contrastive learning can constrain modules to learn unique functions and reduce similarity between them, we use t-SNE[40] to visualize the output embeddings of K(6 in our setting) multiple modules in Figure 5. The visualization of CMTA and CMTA without CL are obtained by multiple embeddings of modules at each time step on MT10-Fixed for 10 episodes. It can be observed that without contrastive learning constraints, the encodings of multiple modules are mixed together.

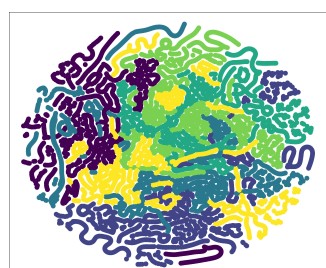
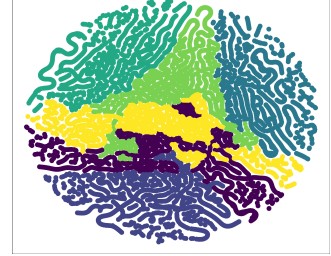

(a) CMTA w/o CL           (b) CMTA

Figure 5: t-SNE visualization of multiple modules' encodings on MT10-Fixed environment.

By using CL, the encodings of different modules become more distinct, indicating that CL indeed reduces the similarity between modules.

# 6 Limitation

One limitation is the computational cost of CMTA, the contrastive learning part is proportional to $O(n^2)$, where n is number of experts. If task similarity is substantial, then even with an increase in the number of tasks, computational cost won't necessarily escalate as long as the number of experts remains constant. Another limitation is the dependence on task similarity, and it's the fundamental assumption in the field of mtrl, even humans struggle to extract mutually beneficial information from entirely unrelated tasks. For instance, attempting to learn from a combination of tasks like playing Go and playing Atrai games would likely yield limited benefits.

# 7 Conclusion

In this paper, we present contrastive modules with temporal attention, which is a generalized way of learning distinct shared modules with finer granularity for multi-task RL. CMTA has the ability to constrain the modules to be different from each other by contrastive learning and the ability to learn the best combination of distinct shared modules dynamically at each time step across tasks by using temporal attention. The module combination in CMTA varies form step to step, so that modules can be reused when it's helpful and not shared when it's conflict, alleviating the negative transfer between within the task and shows great superiority especially for complex tasks. Experiments on Meta-World show that our method outperforms all baselines, both on sample efficiency and performance, and it's the first time achieved performance surpassing that of training each task separately. As the variety of tasks increases(from MT10 to MT50, from Fixed environment to Mixed environment), the advantages of our approach become more apparent. Our method has achieved good results in multi-task RL, and we will further extend it to meta learning in future work.

## Acknowledgements

This work is partially supported by the NSF of China(under Grants 61925208, 62102399, 62222214, 62002338, U19B2019), CAS Project for Young Scientists in Basic Research(YSBR-029), Youth Innovation Promotion Association CAS and Xplore Prize.

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

# A  Pseudo Code

---
**Algorithm 1** CMTA
---
**Initialize**: replay buffer $D$ with $\emptyset$
**Initialize**: initial hidden state $h_0$ with zero tensor
**Initialize**: policy $\pi$ with $\phi$, Q-function Q, task encoder $g$, k experts $f^1, \cdots, f^k$, $lstm$, fully connected layer $\mathcal{W}$
**Input:** state $s_t$ for each environment, one-hot task id $z_\tau$

  1: **for** episode $m = 1, 2, \cdots$ **do**
  2:    **for** time-step $t = 1, 2, \cdots$ **do**
  3:        **for** each task $\tau_i$ **do**
  4:            $z_{enc}^j = f^j(s_t), \forall j \in 1, \cdots, k$
  5:            $z_{task} = g(z_\tau)$
  6:            $h_t = lstm(s_t; h_{t-1})$
  7:            $\alpha_1, \cdots, \alpha_k = softmax(\mathcal{W}(h_t; z_{task}))$
  8:            $z_{enc} = \sum_{j=1}^k \alpha_j \cdot z_{enc}^j$
  9:            $z = z_{task}||z_{enc}$
10:            sample action $a_t \sim \pi(\cdot|z_{task}; z_{enc})$
11:            Perform action $a_t$, get reward $r_t$ and next state $s_{t+1}$.
12:            $D = D \bigcup \{s_t, a_t, r_t, s_{t+1}, h_{t-1}, h_t, z_\tau\}$
13:        **end for**
14:        randomly sample batch from $D$
15:        compute $L_{contrastive}$ by Eq 3
16:        compute $L_{actor}$ and $L_{critic}$ by Eq9 and Eq10
17:        update k experts with $L_{contrastive}$
18:        update $\pi_\phi$ with $L_{actor}$
19:        update all components except $\pi_\phi$ with $L_{critic}$
20:    **end for**
21: **end for**

---

# B  Libraries

We use the following open-source libraries: MetaWorld[3], MTEnv[4],MTRL[5][33].

# C  Smooth factor

$$smoothed\_point[i] = \begin{cases} smoothed\_point[i-1] * factor + point[i] * (1 - factor), & \text{if } i > 0 \\ point[i], & \text{if } i = 0 \end{cases}$$

# D  Hyperparameter Details.

---

[3]https://github.com/rlworkgroup/metaworld, commit-id:af8417bfc82a3e249b4b02156518d775f29eb289
[4]https://github.com/facebookresearch/ mtenv
[5]https://github.com/facebookresearch/mtrl

Table 3: Hyperparameter values that are common across all the methods

| Hyperparameter | Hyperparameter values |
|---|---|
| batch size | 128 × number of tasks |
| network architecture | feedforward network |
| actor/critic size | three fully connected layers with 512 units |
| non-linearity | ReLU |
| policy initialization | standard Gaussian |
| temperature | learned and distangled with tasks |
| exploration parameters | run a uniform exploration policy 1500 steps |
| num of samples / num of train steps per iteration | 1 env step / 1 training step |
| evaluation frequency | 3000 steps |
| replay buffer size | 5000000 |
| policy learning rate | 3e-4 |
| Q function learning rate | 3e-4 |
| optimizer | Adam |
| policy learning rate | 3e-4 |
| beta for Adam optimizer for policy | (0.9, 0.999) |
| Q function learning rate | 3e-4 |
| beta for Adam optimizer for Q function | (0.9, 0.999) |
| discount | 0.99 |
| Episode length (horizon) | 150 |
| reward scale | 1 |

Table 4: Hyperparameter values of task encoder

| Hyperparameter | Hyperparameter values |
|---|---|
| task encoder train from scratch | embedding layer with dim 64 + FC 128 + FC 64 + FC 64 |
| pretrained | pre-trained embedding layer with dim 512 + FC 128 + FC 64 + FC 64 |

Table 5: Hyperparameter values of Soft Modularization

| Hyperparameter | Hyperparameter values |
|---|---|
| task encoder type | train from scratch |
| routing network size | 4 layers and 4 modules per layer with dim 64 |

Table 6: Hyperparameter values of CARE

| Hyperparameter | Hyperparameter values |
|---|---|
| task encoder type | pre-trained embedding layer |
| encoder size | FC 64 + FC 64 |
| number of encoders | 6 |

Table 7: Hyperparameter values of CMTA

| Hyperparameter | Hyperparameter values |
|---|---|
| task encoder type | train from scratch |
| encoder size | FC 64 + FC 64 |
| number of encoders | 6 |
| $\beta$ | 2500 |

# E    Additional Experiment Results

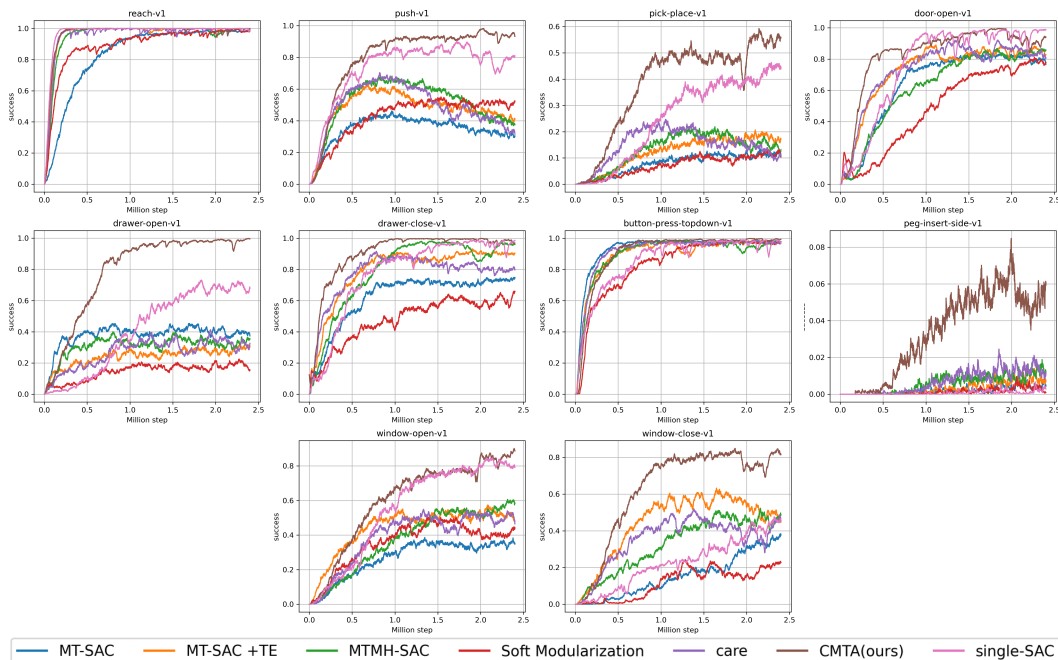

Figure 6: Training curves of different methods on each task of MT10-Mixed, each curve is averaged over 8 seeds. Our approach consistently outperforms baselines in all tasks, whether on asymptotic performance or sample efficiency.

Table 8: Ablations on experts number. Evaluation performance on MT10 tasks after training for 1.0 million steps (for each task). Results have been averaged over 8 seeds.

| agent | MT10-Fixed | | MT10-Mixed | |
|---|---|---|---|---|
| | success rate | | success rate | |
| | max smoothed | max | max smoothed | max |
| CMTA with 2 experts | 54.26% | 62.5% | 52.53% | 57.5% |
| CMTA with 4 experts | 67.68% | 72.5% | 72.19% | 76.25% |
| CMTA with 12 experts | **71.21%** | **75.0%** | 73.18% | 78.75% |
| CMTA with 20 experts | 68.12% | 72.5% | 75.01% | 78.75% |
| CMTA with 6 experts | 68.25% | 73.75% | **78.49%** | **82.5%** |

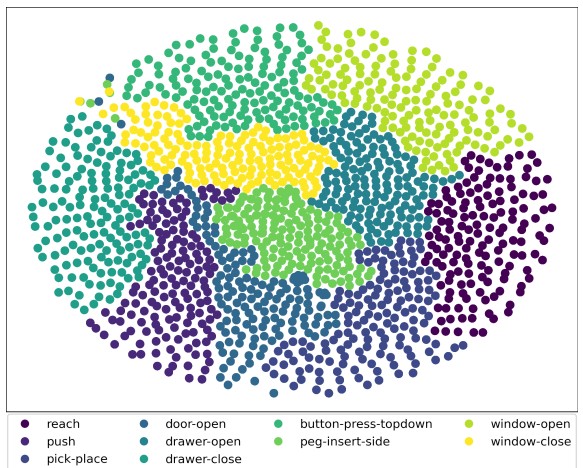

Figure 7: t-SNE visualization of CMTA attention weights on MT10-Mixed environment.

