# OpenReview forum: "Contrastive Modules with Temporal Attention for Multi-Task Reinforcement Learning"
_NeurIPS.cc/2023/Conference — NeurIPS 2023 poster_

### Official Review · Reviewer_sdV7 · 2023-07-05

**Soundness:** 3 good
**Presentation:** 3 good
**Contribution:** 3 good
**Rating:** 7
**Confidence:** 4

**Summary:**

The paper studies modular multi-task reinforcement learning to address negative transfer problem. The proposed method has two components: 1. contrastive learning on module outputs, to encourage model expressiveness and generalization. 2. use temporal information to combine module outputs, to address negative transfer within tasks. Experiments in Meta-World shows that the proposed method outperforms other multi-task baselines and learning tasks individually.

**Strengths:**

1. The motivation for modular learning with temporal attention to address negative transfer is clear.

2. The paper is very well-written, with relevant references and easy-to-follow narration.

3. The method is the only one outperforming single-task RL in Meta-World.

**Weaknesses:**

1. The experiments don't reflect the claim that the method improves generalization.

2. Some previous works (e.g., Multi-Task Reinforcement Learning with Soft Modularization) also select different modules in different time steps in a task.

**Questions:**

1. Why are the results in Figure 3 quite different to the results in "Soft Modularization". In that paper, Soft Modularization outperforms multi-task variants of SAC a lot.

2. Since most multi-task methods cannot outperform the individual-learning baseline, what are the benefits of multi-task RL? Can the learned multi-task model adapt to unseen tasks quicker?

**Limitations:**

The authors have addressed the limitations.

---

> ### Author Rebuttal · Authors · 2023-08-09
>
> We appreciate your valuable feedback and would like to thank you for your time and effort in reviewing our manuscript.  Any further discussion will be appreciated.
>
> > W1: The experiments don't reflect the claim that the method improves generalization.
>
> We believe that the following changes in settings have contributed to increased task complexity and diversity:
>
> 1. The number of tasks increasing from 10 (MT10) to 50 (MT50).
> 2. Transitioning from fixed positions to varying positions (mixed).
>
> In both of these scenarios, our method exhibits greater advantages over other baselines (as indicated in lines 259-262). This to a certain extent underscores the superior generalization ability of our approach.
>
> > W2: Some previous works (e.g., Multi-Task Reinforcement Learning with Soft Modularization) also select different modules in different time steps in a task.
>
> In comparison to SoftModu, our method differs not only in the utilization of LSTM but also in the specifics of attention computation and their application. For instance, while we concatenate temporal information with task information to calculate attention, SoftModu employs a dot product between state information and task information, involving multiple layers of attention weight computation.
>
> > Q1:  Why are the results in Figure 3 quite different to the results in "Soft Modularization". In that paper, Soft Modularization outperforms multi-task variants of SAC a lot.
>
> We mentioned this point in Section 5.4 Lines 277-283. The original SoftModu paper employed a loss weighting trick, which we deliberately omitted in our experiments to ensure fairness in comparison. Additionally, Figure 6(b) in the SoftModu paper (Yang et al., 2020) displays the performance without utilizing this trick, and it aligns quite closely with our reproduction results.
>
> > Q2.1:  Since most multi-task methods cannot outperform the individual-learning baseline, what are the benefits of multi-task RL?
>
> **Better Sample Efficiency:** This advantage can be interpreted as each task benefiting from additional samples generated by auxiliary tasks. Consequently, during the initial training stages, the performance improvement of multi-task rl algorithms outpace that of Single-SAC.
>
> **Model Capacity Compression:** By employing a single network to address all tasks instead of using n separate networks.
>
> > Q2.2: Can the learned multi-task model adapt to unseen tasks quicker?
>
> This outcome hinges on whether the multi-task model merely memorizes multiple tasks or is capable of extracting commonalities among tasks.

---

> > ### Comment · Reviewer_sdV7 · 2023-08-18
> >
> > Thanks for your detailed response!

---

### Official Review · Reviewer_fVjH · 2023-07-05

**Soundness:** 3 good
**Presentation:** 2 fair
**Contribution:** 3 good
**Rating:** 6
**Confidence:** 4

**Summary:**

This work proposes to enhance the expressiveness and generalization capability of the modular methods in multi-task reinforcement learning by applying contrastive loss over different task modules and encode the task related information with a temporal attention module. This work shows by applying both techniques, the proposed CMTA method can learn modules producing different learned embeddings and outperform baselines in different benchmarks.

**Strengths:**

* Different modules of the method are well-motivated, aligned with intuition and described with detail. Details provided for the community to reproduce the results. Using contrastive learning to enforce different module learn different skills is reasonable.
* The visualization of learned encoding of different modules shows the contrastive learning term encourages the different modules to learn skills with different semantic meanings.
* According to the experiment results, the proposed method CMTA outperformed different baselines by a large margin in different settings.
* The overall writing of the work is easy to follow.

**Weaknesses:**

1. Some experiment results are not well-explained:
* According to the Sec 5, the Mixed version of MetaWorld benchmark is supposed to be more difficult than the fixed version.  Why the Single-SAC is performing worse in the Fixed MT10, and it seems the Single-SAC results (in Fixed setting) is much worse than the reported in previous works.
* All baselines perform worse in Mixed (compared with in Fixed), while the proposed method works better in Mixed Version.
*  Why the performance of all methods significantly drops after a certain stage for MT50-Mixed (similar phenomenon did not appear in MT10-Mixed).
2. Some components of the methods could be better ablated like, how does the number of experts affect the performance of the method. Since this work propose to learn more meaningful skills for different experts via contrastive learning, more discussion on this part would make the work stronger.
3. Visualization of the attention weight for different tasks is missing, which could help the audience understand the proposed method.
4. The use of different skill-use is discussed in previous work (Soft Modularization, where the state information and the task information are used at the same time to output the module selection), and no specific training objective (the difference with previous here seems to be the LSTM ) in this work is addressing this issue.

**Questions:**

*  Some explanation regarding the experiment results mentioned in the weakness section would be appreciated.
* Additional visualization regarding the attention weights for different tasks. And since different modules have different semantic meaning (powered by the contrastive learning), some investigation on what kind of skill a specific expert represents would be interesting as well.
* Though this work claims the method works without any loss weighting trick in the optimization, it would be interesting to see some results from that end,

I would raise my rating if the authors could reasonably address part (given the limited time for rebuttal) or full of my concerns.

**Limitations:**

This work propose a general multi-task RL method, in this case, I think no specific potential negative societal impact or similar things should be addressed. As the author indicated, the current method works well in multi-task RL (with a fixed number of tasks), and would like to see some extension in meta learning or open-vocabulary settings in the future.

---

> ### Author Rebuttal · Authors · 2023-08-09
>
> We extend our sincere appreciation to the reviewer for their valuable insights and constructive feedback. Any further discussion will be appreciated.
>
> > W1.1.1: According to the Sec 5, the Mixed version of MetaWorld benchmark is supposed to be more difficult than the fixed version. Why the Single-SAC is performing worse in the Fixed MT10 ?
>
> Mixed version has varying positions. In theory, the agent is more likely to deduce patterns or objectives of tasks (e.g., moving an object to a goal) rather than merely memorizing fixed positions. Thus it is possible for Single-SAC performing better in Miexed MT10.
>
> > W1.1.2: it seems the Single-SAC results (in Fixed setting) is much worse than the reported in previous works.
>
> In the CARE paper, the performance of Single-SAC reached 90%. However, in our reproduction, we could only achieve this level of performance with a single seed. We raised an issue on GitHub regarding our inability to reproduce the Single-SAC results, but the authors have not responded. In the SoftModu paper, after averaging over 3 seeds, the performance of Single-SAC was reported as 78.5%. Considering the difference in seed numbers, this result closely aligns with our experimental findings.
>
> > W1.2:  All baselines perform worse in Mixed (compared with in Fixed), while the proposed method works better in Mixed Version.
>
> This might be indicative of our method's superior generalization ability, enabling it to capture invariant patterns across different positions and thus leading to improved performance. In contrast, other methods in the mixed environment could potentially shift from memorizing a single set of positions (as in Fixed) to remembering 50 sets of positions. This shift could contribute to a decline in performance due to increased complexity and task variation.
>
> > W1.3:   Why the performance of all methods significantly drops after a certain stage for MT50-Mixed (similar phenomenon did not appear in MT10-Mixed).
>
> All algorithms exhibit this phenomenon in the MT50-Mixed environment, and we believe this is an inherent issue with the mixed environment itself.  Most of the baselines experience performance degradation around 0.5 million steps, whereas our method's performance decline begins at 1.5 million steps, highlighting its robustness.  We speculate that the reason for this performance drop lies in the more diverse nature of the MT50-Mixed environment.  Overtraining can lead to model overfitting, causing it to prioritize simpler tasks over more challenging ones.  Our observations of individual task success rates support this, in MT10-mixed, among the 10 tasks, only one task experiences a decline in performance during the later stages(Appendix C, Figure 6, push-v1). However, in MT50-mixed, among the same set of 10 tasks, 5 tasks demonstrate performance drops, and these declines occur earlier in the training process compared to MT10-mixed.  Given that our x-axis represents steps for each task, the training data volume for MT50 is five times that of MT10, which can lead to quicker overtraining issues.
>
> > W2:  Some components of the methods could be better ablated like, how does the number of experts affect the performance of the method.
>
> The ablation of experts number can be seen in the pdf of global response. The experimental results indicate that having fewer experts leads to performance degradation. However, increasing the number of experts beyond a certain point does not yield positive effects.
>
> > W3:  Visualization of the attention weight for different tasks is missing, which could help the audience understand the proposed method.
>
> See  the t-sne visualization of attention weights for different tasks in the pdf of global response.  It is obvious that the attention weights of different tasks are clustered into different clusters, which indicates that CMTA will choose different module combinations for different tasks.
>
> > W4:  The use of different skill-use is discussed in previous work (Soft Modularization, where the state information and the task information are used at the same time to output the module selection), and no specific training objective (the difference with previous here seems to be the LSTM ) in this work is addressing this issue.
>
> In comparison to SoftModu, our method differs not only in the utilization of LSTM but also in the specifics of attention computation and their application. For instance, while we concatenate temporal information with task information to calculate attention, SoftModu employs a dot product between state information and task information, involving multiple layers of attention weight computation.
>
> > Q1: Some explanation regarding the experiment results mentioned in the weakness section would be appreciated.
>
>  See the answer of W1.
>
> > Q2:  Additional visualization regarding the attention weights for different tasks.
>
>  See the answer of W3.
>
> > Q3:  Though this work claims the method works without any loss weighting trick in the optimization, it would be interesting to see some results from that end.
>
> We add a relatively simple loss weighting trick in our method:  let the task weightings $\lambda_i$ (see euqation 1 in our paper)  is propotional to  exp(1/($su_i + \delta$)), where $su_i$ is the current success rate of task i.  The evaluation perfomance(average on 8 seeds) after training 1 million steps is:
>
> |                            | smoothed SR on MT10-Mixed |
> | -------------------------- | ------------------------- |
> | CMTA                       | 78.5                      |
> | CMTA+ naive loss weighting | 73.6                      |
>
>
> Interestingly, the introduction of this trick resulted in a decrease in the performance of our method. This might suggest that the naive loss weighting we attempted may not be effective as initially thought. Additionally, there exist numerous studies on loss weighting in MTRL, all of which can potentially be integrated with our method or the baselines we've utilized.

---

> > ### Comment · Reviewer_fVjH · 2023-08-20
> >
> > The response addressed most of my concerns. I'm raising my score to weak accept

---

### Official Review · Reviewer_GQFL · 2023-07-10

**Soundness:** 2 fair
**Presentation:** 1 poor
**Contribution:** 3 good
**Rating:** 5
**Confidence:** 2

**Summary:**

The paper introduces an approach for multi-task RL. Their approach is similar to CARE, which learns separate encoder modules, but they add a contrastive task loss on top of the encoders. They show this approach outperforms all reported baselines on Meta-World (MT-10 and MT-50) and a variant of Meta-World where the initializations are mixed throughout training.

**Strengths:**

- The proposed approach outperforms all reported baselines and independent training on the metaworld tasks in the fully observed setting.
- The proposed solution is straightforward to implement.

**Weaknesses:**

- A lot of the methods section should be moved to a preliminary section because it is difficult to understand what is novel and not novel. The temporal attention section on L168-180 is one example. Overall I found the organization of the paper confusing.
- There are not enough experimental results to fully validate the method (e.g., relevant baselines like PC-grad). In all of the comparisons to baselines the approach is still fairly overlapping with the error bars of other baselines and the meaning of the error bars is not described anywhere in the text.

- It would help the reader to consolidate the terms. The modules are separately referred to as both experts and modules. I would find it more straightforward if the naming was consistent.

**Questions:**

- On L143 "Notably, it surpasses the performance of learning each task individually for the first time in the Meta-World environment" Is this true in the case of MT10? E.g., Figure 3 of Gradient Surgery for Multi-Task Learning (Yu et. al., 2020).
- What are the error bars in the figures?

---

> ### Author Rebuttal · Authors · 2023-08-09
>
> We wish to thank the reviewer for their thorough review and valuable recommendations that have strengthened our paper.  Any further discussion will be appreciated.
>
> > W1: A lot of the methods section should be moved to a preliminary section because it is difficult to understand what is novel and not novel. The temporal attention section on L168-180 is one example. Overall I found the organization of the paper confusing.
>
> Both subsections 4.1 and 4.2 in our method section is novel and thus doesn't need to be placed in the preliminary section. Our innovation lies in two aspects: contrastive learning and temporal attention. The section on temporal attention (L168-180) is indeed one of our contributions, and it diverges from the previous approach of CARE. While CARE employs task information for attention, our attention module incorporates temporal information as well. It's possible that the reviewer overlooked this distinction, leading to confusion.
>
> > W2.1: There are not enough experimental results to fully validate the method (e.g., relevant baselines like PC-grad).
>
> The experimental results of PC-grad can be compared using Table 1 and Table 3 in the CARE paper (Sodhani et al., 2021). From these tables, it becomes evident that CARE outperforms PC-grad and is indeed a stronger baseline:
>
> |        | MT10              | MT50             |
> | ------ | ----------------- | ---------------- |
> | PCGrad | 0.72  $\pm$ 0.022 | 0.5 $\pm$ 0.017  |
> | CARE   | 0.84 $\pm$ 0.051  | 0.54 $\pm$ 0.031 |
>
> The landscape of existing multi-task RL methods spans multiple orthogonal directions, encompassing architecture design, gradient modulation (including PCgrad), and loss weighting. Methods from different directions can indeed be combined, such as PCgrad + our method, PCgrad + CARE, and so on. However, this fusion is not the primary focus of our study. Consequently, we haven't included PCgrad in our baseline comparisons, as the baselines we have chosen primarily reside in the architecture design direction.
>
> > W2.2:  In all of the comparisons to baselines the approach is still fairly overlapping with the error bars of other baselines and the meaning of the error bars is not described anywhere in the text.
>
> Not mentioning the significance of error bars in the paper was indeed an oversight on our part, and we appreciate the reviewer's observation. The error bars (shaded areas) represent the standard deviation across 8 different seeds. In RL experiments, results can be highly influenced by the choice of seeds, making it necessary to average the outcomes across multiple seeds to mitigate the impact of randomness-induced errors. The overlapping error bars among different baselines merely indicate the substantial effect of RL's randomness in MT10. As a result, our primary mode of comparison relies on the mean performance of different algorithms.  Furthermore, in the MT50, there is no overlap among different baselines. This distinct separation clearly highlights the superiority of our approach.
>
> > W3: It would help the reader to consolidate the terms. The modules are separately referred to as both experts and modules. I would find it more straightforward if the naming was consistent.
>
> That's a great suggestion, and we will certainly rephrase the relevant sections in the forthcoming revised version to minimize reader confusion. We appreciate the reviewer's valuable advice.
>
> > Q1: On L143 "Notably, it surpasses the performance of learning each task individually for the first time in the Meta-World environment" Is this true in the case of MT10? E.g., Figure 3 of Gradient Surgery for Multi-Task Learning (Yu et. al., 2020).
>
> In Figure 3 of the PC-grad paper (Yu et al., 2020), PC-grad appears to achieve performance close to that of single task training. However, the paper does not explicitly mention averaging results across multiple seeds in the experiments. Hence, we speculate that PC-grad might have utilized only a single seed for experiments on Meta-World.  The substantial randomness inherent in reinforcement learning, as discussed in W2.2, can weaken the persuasiveness of results from single seed. Based on our experimental findings, it is evident that only our approach outperforms single task training in both MT10-fixed and MT10-mixed.
>
> > Q2: What are the error bars in the figures?
>
> See the answer of W2.2.

---

### Official Review · Reviewer_4rda · 2023-07-10

**Soundness:** 3 good
**Presentation:** 2 fair
**Contribution:** 3 good
**Rating:** 5
**Confidence:** 4

**Summary:**

This paper proposes an approach to multi-task RL called Contrastive Modules with Temporal Attention that aims to address the issue of negative transfer between tasks in multi-task RL.

The proposed method consists of two main components: contrastive learning and temporal attention. The contrastive learning component is used to ensure that the shared modules learned by the method are distinct from each other. This is achieved by applying a contrastive loss that encourages the modules to produce different outputs for the same input. The temporal attention component is used to dynamically combine the outputs of the different modules at each time step. This allows the method to adapt to the specific requirements of each task.

The authors evaluate their method on the Meta-World benchmark, a widely used benchmark for multi-task RL. They compare the performance of their method with several baselines, including methods that train each task separately and methods that share all modules across tasks. The results show that CMTA outperforms the baselines in terms of both sample efficiency and performance.

**Strengths:**

Originality
- The paper presents a novel approach to multi-task RL by introducing contrastive modules with temporal attention.
- The method addresses the issue of negative transfer between tasks, which is a significant challenge in multi-task RL. The authors propose a novel solution to this problem by constraining the modules to be different from each other and using temporal attention to dynamically combine them.

Quality
- The authors provide adequate experimental results on Meta-World, a widely-accepted continuous control robotics benchmark, and some ablation studies that support the effectiveness of their method.
- The paper is well-referenced, indicating a thorough understanding of the existing literature. The authors clearly position their work within the context of previous research.

Clarity:
- The paper is well-organized and the writing is clear. The authors provide a clear explanation of their method and its advantages.
- The figures and tables in the paper are informative and support the text well. They help to clarify the method and the experimental results.

Significance:
- The proposed method addresses a critical challenge in the field and shows superior performance compared to existing methods.
- The method proposed by the authors, in particular the temporal attention mechanism, has the potential to be widely adopted in the field of multi-task RL. It could also inspire future research in this area.

**Weaknesses:**

Limited insight into hyperparameter sensitivity: The paper would be stronger if it discussed the sensitivity of the proposed method to its hyperparameters. Understanding how changes in hyperparameters affect the performance of the model is crucial for reproducibility and for users who wish to apply the method to their own tasks.

Lacking insight into soft attention weights: It would be interesting to see the general relationships of soft attention weights and other aspects of the problem, such as its temporal nature, the tasks involved, etc.

Limited Discussion on Failure Cases: While the paper presents a number of successful results, it could discuss in detail the scenarios where the proposed method fails or performs sub-optimally. Such a discussion could provide valuable insights into the limitations of the method and guide future improvements.

Lack of Comparison with Related Work: While the paper compares the proposed method with several baselines, it does not compare it with other multiple other methods that also use contrastive learning or attention mechanisms in the context of multi-task RL (*see first bullet point below). Such comparisons could provide a more comprehensive evaluation of the proposed method.
  - However, the paper does add the contrastive loss to CARE and evaluate it, but CARE's attention mechanism is very different than CMTA's and only CARE was compared against in this way.
  - It would be interesting to see how other proposed baselines or competitive related work would benefit from the proposed temporal attention module.

Generalizability: The proposed method has been evaluated on a specific benchmark (Meta-World). Its performance on other benchmarks or real-world tasks that differ from Meta-World type setup and tasks is not known. CMTA may not outperform certain baselines or other algorithms on suites of tasks in other benchmarks or real-world tasks.

**Questions:**

Subsumed into Weaknesses and Limitations.

1. What are three limitations of your work that have not been addressed in the reviews, and what are your thoughts about them? This isn't intended to diminish your work. Instead it's to show that you understand where and how your work shines and to highlight where it may not as future problems to be addressed or as problems that are insignificant for some reason(s).

**Limitations:**

Scalability: The proposed method might not scale well to tasks with a larger number of subtasks or more complex environments, even though the paper claims in Line 311-312 that "as the variety of tasks increases, the advantages of our approach become more apparent", they don't show this outside of Meta-World, and Meta-World is measure 0 subset of possible tasks.

Regardless, the computational cost of the method, especially the contrastive learning part, could increase significantly with the complexity of the tasks.

Dependence on Task Similarity: The effectiveness of the proposed method might depend on the similarity of the tasks. If the tasks are very different from each other, the shared modules learned by the method might not be effective for all tasks.

I did not find the paper addressing its limitations anywhere. One I will put forth is the lack of insight into the temporal attention module's relationship to different aspects of the problem, such as (1) how the attention weights vary over time, (2) whether they reach a steady state, (3) what do they focus on, for the Meta-World problems covered in the paper.

I believe we should be wary of how this overall approach behaves, especially the temporal attention module and the assigned soft attention weights, in problems in which ethics, safety, fairness, bias, may be a concern. This is general concern for any algorithm that doesn't explicitly address and mitigate these issues, but it's especially one here because of the lack of insight into the temporal attention module's relationship to aspects of the problem.

---

> ### Author Rebuttal · Authors · 2023-08-09
>
> We are grateful to the reviewer for their thoughtful suggestions and comments that have greatly improved our manuscript.  Any further discussion will be appreciated.
>
> > W1: Limited insight into hyperparameter sensitivity
>
> The ablation of experts number can be seen in the pdf of global response. The experimental results indicate that having fewer experts leads to performance degradation. However, increasing the number of experts beyond a certain point does not yield positive effects.
>
> >  W2:  Lacking insight into soft attention weights: It would be interesting to see the general relationships of soft attention weights and other aspects of the problem, such as its temporal nature, the tasks involved, etc.
>
> - temporal nature: see the answer of L4
> - tasks involved：see the t-sne visualization of attention weights for different tasks in the pdf of global response.  It is obvious that the attention weights of different tasks are clustered into different clusters, which indicates that CMTA will choose different module combinations for different tasks.
>
> > W3: Limited Discussion on Failure Cases
>
> Some discussion on failure cases including:
>
> - According to the Sec 5, the Mixed version of MetaWorld benchmark is supposed to be more difficult than the fixed version. Why the Single-SAC is performing worse in the Fixed MT10, and it seems the Single-SAC results (in Fixed setting) is much worse than the reported in previous works.
> - All baselines perform worse in Mixed (compared with in Fixed), while the proposed method works better in Mixed Version.
> - Why the performance of all methods significantly drops after a certain stage for MT50-Mixed (similar phenomenon did not appear in MT10-Mixed).
>
>  Due to character limitations, the answers can be found in reviewer fVjH's W1.
>
> > W4: Lack of Comparison with Related Work
>
> Currently, there are no other MTRL algorithms utilizing contrastive learning. Additionally, the employment of attention mechanisms is limited to the CARE and SoftModu. Our use of contrastive learning and temporal attention is built upon a mixed-of-experts (MOE) design, which restricts their applicability to methods utilizing MOE (specifically, only CARE employs MOE). To validate the temporal attention module on other baselines like MTSAC, it would indeed necessitate the introduction of an MOE. However, this modification would alter the original structure (e.g., SAC + temporal attention (+MOE) = CMTA w/o CL). In essence, this corresponds to a portion of our ablation experiments.
>
> > W5: Generalizability: The proposed method has been evaluated on a specific benchmark (Meta-World). Its performance on other benchmarks or real-world tasks that differ from Meta-World type setup and tasks is not known. CMTA may not outperform certain baselines or other algorithms on suites of tasks in other benchmarks or real-world tasks.
>
> MetaWorld provides a diverse task distribution with 50 different tasks involving objects like doors, cups, windows, drawers, etc. and skills like push, pull, open, close, etc.  Evaluating on a broad task distribution（Metaworld） provides a good estimate of the generalization capabilities of mtrl algorithms .
>
> Currently, MetaWorld stands as the sole widely recognized benchmark in mtrl, and previous mtrl works (CARE(Sodhani et al., 2021), SoftModu(Yang et al., 2020)) also  only evaluate their methods on MetaWorld. Therefore, we need to first construct benchmarks before implementing the algorithm. So, we will leave it as future work.
>
> > L1: Scalability
>
> As mentioned in W5,  MetaWorld stands as the sole widely recognized benchmark in mtrl and provides a good estimate of the generalization capabilities of mtrl algorithms.  In Meta-World,  we believe that the following changes in settings have contributed to increased task complexity and diversity:
>
> - The number of tasks increasing from 10 (MT10) to 50 (MT50).
>
> - Transitioning from fixed positions to varying positions (mixed).
>
> In both of these scenarios, our method exhibits greater advantages over other baselines (as indicated in lines 259-262). This outcome to some extent substantiates the claim that "as the variety of tasks increases, the advantages of our approach become more apparent."
>
> > L2:   Regardless, the computational cost of the method, especially the contrastive learning part, could increase significantly with the complexity of the tasks.
>
> The computational cost of the contrastive learning part is proportional to $O(n^2)$ , where n is number of experts.  If task similarity is substantial, then even with an increase in the number of tasks, computational cost won't necessarily escalate as long as the number of experts remains constant. For instance, in the case of MT50, we employed the same six experts as in MT10.
>
> > L3:   Dependence on Task Similarity
>
> Task similarity is indeed a fundamental aspect of multi-task learning. Even humans struggle to extract mutually beneficial information from entirely unrelated tasks. For instance, attempting to learn from a combination of tasks like playing Go and play atrai games would likely yield limited benefits.
>
> > L4:  I did not find the paper addressing its limitations anywhere. One I will put forth is the lack of insight into the temporal attention module's relationship to different aspects of the problem.
>
> Based on our observations, the trends in attention weights exhibit rapid changes during the early stages of each episode, followed by continuous fluctuations within a narrow range. It is plausible that the agent rapidly infers the necessary skills during the initial phase and subsequently refines the skills it employs based on the discrepancies between execution and prediction. Since our attention mechanism does not directly operate on input observations, it becomes challenging to directly infer what specific aspects the agent is focusing on.

---

> > ### Comment · Reviewer_4rda · 2023-08-19
> >
> > Thank you for your rebuttal. I've read it, the global pdf, and the other reviews and associated rebuttal discussions.
> >
> > W2.
> > I'm not a fan of only providing t-SNE because of utter lack of parameter invariance in qualitative insights. If using sklearn, the documentation [1] shows that the function has the following parameters,
> >
> > class sklearn.manifold.TSNE(n_components=2, *, perplexity=30.0, early_exaggeration=12.0, learning_rate='auto', n_iter=1000, n_iter_without_progress=300, min_grad_norm=1e-07, metric='euclidean', metric_params=None, init='pca', verbose=0, random_state=None, method='barnes_hut', angle=0.5, n_jobs=None)
> >
> > and parameters, such as perplexity, that greatly affect the visualization are not varied or expressly detailed.
> >
> > Regardless, the clusters are distinct though the paper would be stronger if a reasonable attempt at reproducibility of the t-SNE plots were made possible. It may not be possible to make it completely reproducible, as t-SNE has a non-convex loss function. However, if the only difference is randomization, then with such nice clusters, there should be no real issue getting nice clusters again unless there was an issue acquired nice clusters originally. I'd suggest providing as much information as possible on your process for creating t-SNE plots, as a general practice since it is little effort, and pairing t-SNE with a data visualization method that is less user-manipulatable.
> >
> > W4.
> > Regarding works using contrastive learning in MTRL. I apologize, I was over-reaching by phrasing it this way. Instead this work is MTRL using contrastively learned MoE embeddings + context task embeddings. So for learning distinct experts in MoE setting, there are continual learning methods that perform OoD detection prior to creating a new task [3, 4] and/or use similarity measures such as a Kullback-Leibler, Jensen-Shannon loss or Wasserstein distance loss to encourage dissimilarity between experts.
> >
> > There are also approaches that use contrastive learning on MoEs to learn distinct experts [5].
> >
> > I don't grasp the point(s) being made in the remainder of this paragraph that is speaking to attention mechanisms.
> >
> > W5.
> > Meta-World generalization provides a good estimate for generalization on unseen Meta-World and similar tasks, which are a tiny space of continuous control tasks that are similar to the seen Meta-World tasks.
> >
> > For other items, I don't have any further comments. Thank you for your work and rebuttal. I will maintain my score as is and encourage the Authors' to consider the Reviewers' feedback and rebuttal discussions in improving their paper.
> >
> > [1] https://scikit-learn.org/stable/modules/generated/sklearn.manifold.TSNE.html.
> > [2] Eysenbach, B., Zhang, T., Levine, S., & Salakhutdinov, R. R. (2022). Contrastive learning as goal-conditioned reinforcement learning. Advances in Neural Information Processing Systems, 35, 35603-35620.
> > [3] Nagabandi, A., Finn, C., & Levine, S. (2018). Deep online learning via meta-learning: Continual adaptation for model-based rl. arXiv preprint arXiv:1812.07671.
> > [4] Xu, M., Ganesh, S., & Pasula, P. (2022). Mixture of basis for interpretable continual learning with distribution shifts. arXiv preprint arXiv:2201.01853.
> > [5] Mustafa, B., Riquelme, C., Puigcerver, J., Jenatton, R., & Houlsby, N. (2022). Multimodal contrastive learning with limoe: the language-image mixture of experts. Advances in Neural Information Processing Systems, 35, 9564-9576.

---

> > > ### Author Response · Authors · 2023-08-21
> > >
> > > Thank you for your response. I have read the comment, including the papers you mentioned [3, 4, 5].
> > >
> > > W2.
> > >
> > > We use the following instruction to get t-SNE visualization and we will add this to the appendix in the revised version:
> > >
> > > ```
> > > tsne = sklearn.manifold.TSNE(n_components=2, init ='pca', random_state=40)
> > > ```
> > >
> > > For other parameters that might affect the visualization, we used the default values.  Furthermore, we have also altered the `random_state` to visualize the data again, and we still obtained nice clusters.
> > >
> > > W4.
> > >
> > > In [3, 4], apart from employing the mixture of models, there seems to be no explicit demonstration of constraining dissimilarity between modules, which presents a significant contrast with our approach. While in other domains, the use of similarity measures to encourage dissimilarity between experts does indeed share a similar motivation with our work, the implementation methods differ from ours.
> > >
> > > As for [5], it belongs to the domain of multi-modal. The purpose of utilizing contrastive learning is to align image and text representations, treating  corresponding $Z_{text}$ and $Z_{image}$ as positive pairs. So it cannot be directly applied to MTRL as its purpose and implementation details are distinct from ours(we use the output of current time-step and next time-step of the same expert as postive pairs instead).
> > >
> > > W5.
> > >
> > > Indeed, Meta-World tasks do exhibit certain limitations; However, at present, there is no superior benchmark available for MTRL.  In the future, when new benchmarks become available, it might be possible to further validate our approach on them.
> > >
> > > [3] Nagabandi, A., Finn, C., & Levine, S. (2018). Deep online learning via meta-learning: Continual adaptation for model-based rl. arXiv preprint arXiv:1812.07671.
> > >
> > > [4] Xu, M., Ganesh, S., & Pasula, P. (2022). Mixture of basis for interpretable continual learning with distribution shifts. arXiv preprint arXiv:2201.01853.
> > >
> > > [5] Mustafa, B., Riquelme, C., Puigcerver, J., Jenatton, R., & Houlsby, N. (2022). Multimodal contrastive learning with limoe: the language-image mixture of experts. Advances in Neural Information Processing Systems, 35, 9564-9576.

---

### Official Review · Reviewer_ouGh · 2023-07-24

**Soundness:** 2 fair
**Presentation:** 2 fair
**Contribution:** 2 fair
**Rating:** 5
**Confidence:** 5

**Summary:**

The paper focuses on multi-task reinforcement learning. Motivated by the negative task transfer within each task, the paper proposes the Contrastive Modules with Temporal Attention (CMTA) method, which utilizes temporal attention to modulate the weights of experts. Concretely, temporal attention takes recent history as input to capture local information. The proposed CMTA is evaluated in MetaWorld and shows superior performance than baselines.

**Strengths:**

- The methodology part is clear and well-written.

 - The methodology is well-motivated.

**Weaknesses:**

- The paper poses the multiple skills that may be utilized in each task as negative transfer problems within a task. I believe such terminology is ok but I would highly recommend including a session in related work summarizing the skill-based RL literature and pointing out the deep connection between “negative transfer within a task” and “skill-based RL” in the introduction part.

 - The novelty of the paper is limited from my point of view.

 - The experiment part lacks necessary information and please refer to the questions. The ablation study can be further improved by analyzing the impact of history length in the temporal attention mechanism, as well as the number of experts.

**Questions:**

- What is the number of experts in the mixture for each experiment setting, including MT10-Fixed, MT10-Mixed, MT50-Fixed, MT50-Mixed?

 - How is the smooth curve calculated exactly? The meaning of the smooth factor in line 242 is unclear.

 - Why do the performances drop for MT50-Mixed after 1.5 Million steps (figure 3 the right most figure MT50-Mixed)?

**Limitations:**

- The limitation is not discussed in the conclusion. One trade-off is the increased model size due to experts and the improved performance.

---

> ### Author Rebuttal · Authors · 2023-08-09
>
> Our gratitude goes to the reviewer for their insightful comments, which have significantly enhanced the quality of our work.  Any further discussion will be appreciated.
>
> > W1:  The paper poses the multiple skills that may be utilized in each task as negative transfer problems within a task. I believe such terminology is ok but I would highly recommend including a session in related work summarizing the skill-based RL literature and pointing out the deep connection between “negative transfer within a task” and “skill-based RL” in the introduction part.
>
> We appreciate the reviewer's suggestions. We think that "skill-based RL" indirectly addresses the issue of "negative transfer within a task" by employing distinct skills. However, its primary focus lies in using hierarchical reinforcement learning to tackle intricate tasks that are challenging for a single policy, or inducing diverse skills through reward shaping during pretraining to adapt swiftly to new tasks. We will incorporate this analysis in the forthcoming revised version of the paper.
>
> > w2:  The novelty of the paper is limited from my point of view.
>
> The novelty of this paper is primarily twofold: contrastive learning and temporal attention. These address two aspects of non-modularity in current  MTRL methods. Both of these aspects were acknowledged in the "Originality" by Reviewer 4rda. In the paper, we conducted ablation experiments on the introduced components individually (as shown in Table 2 and Figure 4). These experiments demonstrated the effectiveness of both components, resulting in performance improvements of approximately 5% and 20%, respectively. Based on the summary part, it appears that the reviewer might have mainly focused on the second aspect, potentially leading to confusion regarding the novelty of the paper.
>
> > W3： The experiment part lacks necessary information and please refer to the questions. The ablation study can be further improved by analyzing the impact of history length in the temporal attention mechanism, as well as the number of experts.
>
> - number of experts:  The ablation result can be seen in the pdf of global response. The experimental results indicate that having fewer experts leads to performance degradation. However, increasing the number of experts beyond a certain point does not yield positive effects.
>
> - history length: During the data collection process, we store both the current hidden state and the next hidden state . Consequently, each sample comprises the elements (s, a, r, s', h, h'). This structure enables our temporal attention mechanism to effectively encompass the historical information of the entire trajectory up to the current time step. Notably, the history length is not a hyperparameter here, which makes conducting ablation experiments on it potentially unnecessary.
>
> > Q1: What is the number of experts in the mixture for each experiment setting, including MT10-Fixed, MT10-Mixed, MT50-Fixed, MT50-Mixed?
>
> We use 6 experts for all settings, as mentioned in the Appendix D.
>
> > Q2: How is the smooth curve calculated exactly? The meaning of the smooth factor in line 242 is unclear.
>
> Thanks for bringing this omission to our attention. We will certainly address this in the revised version by including the calculation method for the smooth curve:
>
> $$smoothed\\_point[i] =  \\begin{cases} smoothed\\_point[i-1]* factor + point[i] *(1-factor), & \\text{if $i$ > 0} \\\\  point[i], & \\text{if $i$ = 0} \\end{cases}$$
>
> > Q3: Why do the performances drop for MT50-Mixed after 1.5 Million steps (figure 3 the right most figure MT50-Mixed)?
>
> All algorithms exhibit this phenomenon in the MT50-Mixed environment, and we believe this is an inherent issue with the mixed environment itself.  Most of the baselines experience performance degradation around 0.5 million steps, whereas our method's performance decline begins at 1.5 million steps, highlighting its robustness.  We speculate that the reason for this performance drop lies in the more diverse nature of the MT50-Mixed environment.  Overtraining can lead to model overfitting, causing it to prioritize simpler tasks over more challenging ones.  Our observations of individual task success rates support this, as some tasks indeed show declines in success rates during later stages. In MT10-mixed, among the 10 tasks, only one task experiences a decline in performance during the later stages(Appendix C, Figure 6, push-v1). However, in MT50-mixed, among the same set of 10 tasks, 5 tasks demonstrate performance drops, and these declines occur earlier in the training process compared to MT10-mixed.  Given that our x-axis represents steps for each task, the training data volume for MT50 is five times that of MT10, which can lead to quicker overtraining issues.

---

> > ### Comment · Reviewer_ouGh · 2023-08-20
> > **Rebuttal Response**
> >
> > I thank the authors for the detailed response, which helped improve the paper's clarity. I updated my score accordingly.

---

### Author Rebuttal · Authors · 2023-08-09

The PDF here includes our ablation experiments on the number of experts and the t-SNE visualization of CMTA attention weights of different tasks.

---

### Author Response · Authors · 2023-08-18
**Dear reviewers**

 Note that the discussion will be ended in 3 days, we kindly request all the reviewers engage in the discussion. During the rebuttal period,we carried out additional experiments and are here to ensure that we have addressed all questions or doubts . We hope everything is clear and the recent experiments have addressed the concerns. Thanks!

---

### Decision · Program_Chairs · 2023-09-21

**Decision:**

Accept (poster)

**Comment:**

The paper presents a novel solution to the negative transfer problem in multi-task reinforcement learning. It introduces the CMTA method, which employs contrastive learning to ensure diverse modules and uses temporal attention to combine them effectively. The paper's strengths include its novel approach, compelling experimental results outperforming benchmarks, simple implementation, and clear presentation. The author should aim to incorporate the suggestions put forward by reviewers, in particular, those around discussing limitations, additional visualisations, and clarifications.